# Evaluation of In Vitro Wound-Healing Potential, Antioxidant Capacity, and Antimicrobial Activity of *Stellaria media* (L.) Vill

Florina Miere (Groza) [1,†], Alin Cristian Teușdea [2,*,†], Vasile Laslo [2,†], Simona Cavalu [1,*,†], Luminița Fritea [1,†], Luciana Dobjanschi [1,†], Mihaela Zdrinca [1,†], Marcel Zdrinca [1,†], Mariana Ganea [1,†], Priscilla Pașc [1,†], Adriana Ramona Memete [3,†], Angela Antonescu [1,†], Andreea Margareta Vlad [1,†] and Simona Ioana Vicas [2,*,†]

1 Faculty of Medicine and Pharmacy, University of Oradea, 10 P-ta 1 December Street, 410073 Oradea, Romania; florinamiere@uoradea.ro (F.M.); lfritea@uoradea.ro (L.F.); dobjanschil@uoradea.ro (L.D.); mzdrinca@uoradea.ro (M.Z.); marcelzdrinca@yahoo.ro (M.Z.); mganea@uoradea.ro (M.G.); priscilla_pasc@uoradea.ro (P.P.); angela.antonescu@didactic.uoradea.ro (A.A.); dalvnoi@yahoo.com (A.M.V.)
2 Faculty of Environmental Protection, University of Oradea, 26 Gen. Magheru Street, 410048 Oradea, Romania; vasilelaslo@yahoo.com
3 Doctoral School of Biomedical Science, University of Oradea, 10 P-ta 1 December Street, 410073 Oradea, Romania; adrianamemete@yahoo.com
* Correspondence: ateusdea@yahoo.co.uk (A.C.T.); scavalu@uoradea.ro (S.C.); svicas@uoradea.ro (S.I.V.)
† All authors have equally contributed to this paper.

**Abstract:** The healing of skin wounds remains an important concern in medicine, especially in chronic wounds caused by various diseases such as diabetes. Using herbs or herbal products to heal skin wounds is a therapeutic challenge for traditional medicine. In this context, the main aim of our work was to highlight the in vitro healing potential of *Stellaria media* (L.) Vill. (SM) extract using the scratch assay on normal human dermal fibroblasts (NHDF). The ability to stimulate cell migration and proliferation under the influence of different concentrations of SM extract (range between 12.5 and 200 μg/mL) was determined compared to the control (untreated *in vitro*-simulated wound) and positive control (allantoin 50 μg/mL). Our results showed that the concentration of 100 μg/mL SM extract applied on the simulated wound recorded the strongest and fastest (24 h) migration (with wound closure) and proliferation of NHDF compared with the control. In addition, the SM extract was characterized in terms of bioactive compounds (total phenols and flavonoids content), antioxidant capacity (FRAP (The Ferric-Reducing Antioxidant Power) assay and electrochemical method), and antimicrobial activity. The results show that the SM extract contains a considerable amount of polyphenols (17.19 ± 1.32 mg GAE/g dw and 7.28 ± 1.18 mg QE/g dw for total phenol and flavonoid content, respectively) with antioxidant capacity. Antimicrobial activity against Gram-positive bacteria (*S. aureus*) is higher than *E. coli* at a dose of 15 μg/mL. This study showed that *Stellaria media* is a source of polyphenols compounds with antioxidant capacity, and for the first time, its wound healing potential was emphasized.

**Keywords:** *Stellaria media* (L.) Vill; scratch assay; fibroblasts; wound healing; antioxidant capacity; phenols and flavonoids content; antimicrobial activity

## 1. Introduction

The skin is considered the largest and most complex organ of the human body. A major role in maintaining healthy skin is played by dermal fibroblasts, with the ability to synthesize collagen [1]. At the same time, fibroblasts are cells that play a role in maintaining the microbiome and skin immunity (phagocytose aggregate collagen, synthesize interferon-β, and cause the production of interleukins IL-6, IL-1, and IL-8) [2]. Fibroblasts are responsible for the production of acidic sweat, thus creating an unfavorable environment for microorganisms [3]. In addition to the roles mentioned above, the dermal fibroblast is also responsible for the production of extracellular matrices, but also for differentiation under different cell types: reticular and papillary fibroblasts, and intradermal adipocytes [4].

In the case of a lesion, healing takes place in three complex phases (Figure 1).

**Figure 1.** Stages of wound healing and the implications of fibroblasts in this process.

According to Figure 1, the first phase represents the moment of inflammation and cessation of bleeding, involving processes such as vasoconstriction, platelet aggregation, and subsequent coagulation of the aseptic wound by phagocytosis and debridement (it can take up to five days depending on the treatment applied). The second phase represents the actual healing, being the stage of cell proliferation. In this phase, the fibroblasts determine the synthesis of collagen, which "fills" the wound, and then the formation of angiogenesis takes place. Subsequently, the size of the wound decreases due to the contraction process—the collagen fibers are reorganized so that later, the epithelialization takes place (this process requires between 2 and 10 days depending on the treatment applied). The last phase is the remodeling phase, when the collagen synthesis is low but the collagen fibers generated by fibroblasts are firmer and more elastic, and the skin regains its elasticity [5].

The classic therapy used to heal wounds involves various synthetic drugs (antibiotics and antifungals) that, over time, can cause tolerance, with the need to increase the dose applied to decrease the sensitivity of microorganisms to these substances and thus the appearance of infections [6–8]. In general, adult dermal cells cannot migrate and proliferate in the case of injury without leaving traces (scars), but young papillary fibroblasts have this potential [3]. The interest of research [9–11] is focused on identifying new plant sources that stimulate the development and proliferation of fibroblasts [12,13].

Our study investigated the wound-healing effect in vitro by SM extract in NHDF cells using a scratch assay. The rate of migration and proliferation inside the wound of fibroblast cells were quantitatively evaluated by complex statistical analysis. Microscope images of fibroblast cells subjected to various SM extract doses were processed in order to determine the width of the wound, the area of the wound, and the density of cells inside the wound. In addition, SM extract was characterized in terms of bioactive compounds (total phenols and flavonoids content), antioxidant capacity by FRAP assay and an electrochemical method, and antimicrobial activity. To the best of our knowledge, this is the first study to highlight the wound-healing potential of SM extract.

## 2. Results

### 2.1. Total Phenols, Flavonoids Content, and Antioxidant Capacity of SM

The content of bioactive compounds, such as total phenols and flavonoids, along with the antioxidant capacity of *SM* extract, are shown in Table 1.

**Table 1.** Total phenols, flavonoids content, and antioxidant capacity (FRAP assay and electrochemical method) of SM extract *.

| Total Phenols (mg GAE/g dw) | Flavonoids (mg QE/g dw) | FRAP [1] (µg AAE/mL) | AEAC [2] (µg/mL) |
|---|---|---|---|
| $17.19 \pm 1.32$ | $7.28 \pm 1.180$ | $5.54 \pm 0.49$ | $26.00 \pm 0.02$ |

* The results are presented as mean $\pm$ SD (standard deviation). [1] Ferric-Reducing Antioxidant Power (FRAP); [2] Ascorbic Acid Equivalent Antioxidant Capacity (AEAC); Gallic acid equivalent (GAE); Quercetin Equivalent (QE); AAE—Ascorbic Acid Equivalent.

The antioxidant capacity of SM was evaluated by both a spectrophotometric (FRAP assay) and electrochemical method, respectively. The curve obtained by the electrochemical method for antioxidant capacity and the highlighted peaks are shown in Figure 2.

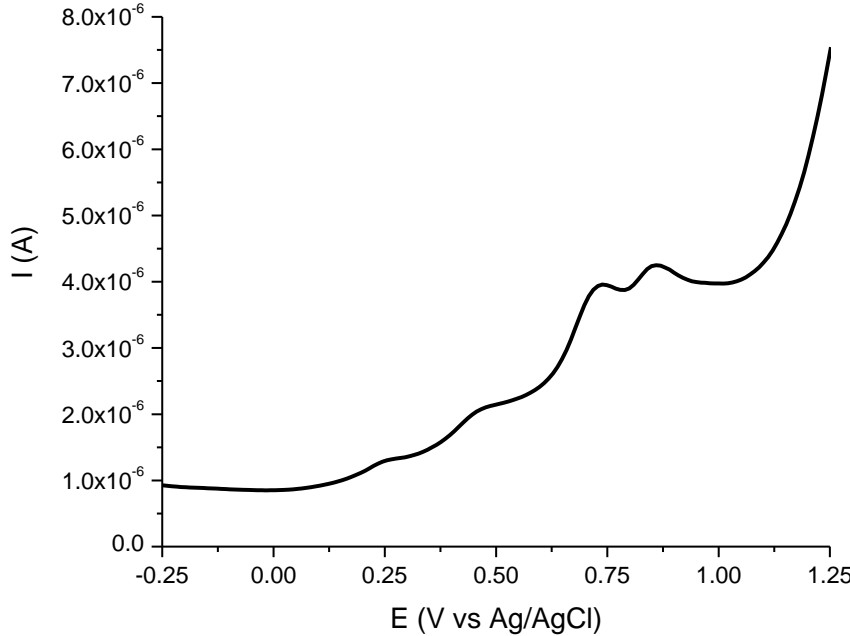

**Figure 2.** Differential pulse voltammogram of the SM extract (1 mg/mL).

The extract was analyzed by differential pulse voltammetry (DPV) (Figure 2), revealing the oxidation process of the chemical compounds with a reducing character from the extract composition. A total of four peaks can be observed (two small shoulders and two better-defined peaks) at different potentials (at around 0.250 V, 0.450 V, 0.700 V, and 0.850 V, respectively) and with different intensities. The peak from around 0.450 V can be attributed to ascorbic acid oxidation, leading to a content of $0.47 \pm 0.003$ µg/mL. The value of AEAC was $26 \pm 0.02$ µg/mL, which indicates an important antioxidant capacity of SM extract, in accordance with other studies employing spectrophotometric methods [14–17].

In the FRAP assay, an electron-transfer-based method, the reduction in the ferric ion-ligand complex to the blue-colored ferrous complex by SM extract was measured. The result was expressed as Ascorbic Acid Equivalent to compare it with the electrochemical method.

### 2.2. Antimicrobial Activity of the SM Extract

The SM extract was investigated to evaluate its antibacterial activity against Gram-positive (*S. aureus*) and Gram-negative (*E. coli*) bacteria using the disc diffusion method. An

evaluation of the antibacterial activity of different concentrations of SM extract is recorded in Table 2.

**Table 2.** Antimicrobial activity against Gram-negative (*E. coli*) and Gram-positive (*S. aureus*) *bacteria* of the different concentrations of SM extract (5 µg/mL, 10 µg/mL, and 15 µg/mL) compared to the following antibiotics: 10U Penicillin (P), 30 µg Doxycycline (DOX), 10 µg Gentamicin (GN), 30 µg Ceftazidime (CAZ), 300 µg Nitrofurantoin (F), 5 µg Ciprofloxacin (CIP), and 30 µg Oxacillin (FOX).

| Antibiotics/Extract | Inhibition Zone (mm) | |
|---|---|---|
| | *E. coli* | *S. aureus* |
| CAZ | 28.33 ± 1.43 | n.d. |
| FOX | n.d. | 35.66 ± 2.30 |
| P | n.d. | 34.33 ± 1.93 |
| CIP | 30.66 ± 2.12 | 27.33 ± 2.01 |
| GN | 20.66 ± 1.45 | 23.66 ± 1.99 |
| DOX | 20.66 ± 1.62 | n.d. |
| F | 22.66 ± 1.23 | 24.33 ± 2.14 |
| SM extract (5 µg/mL) | 7.66 ± 0.85 | 10.66 ± 0.86 |
| SM extract (10 µg/mL) | 8.33 ± 0.56 | 11.66 ± 0.99 |
| SM extract (15 µg/mL) | 9.66 ± 0.57 | 15.33 ± 1.27 |

n.d. = not detected; Data are means of three replicates ($n = 3$) ± standard deviation.

### 2.3. Cell Viability Assay

The effect of different SM extract concentrations, ranging between 12.5 µg/ML and 200 µg/mL on cell viability after incubation for 24 h, was investigated (Figure 3). The percentage of cell viability did not change significantly compared to the control (CTRL0) except for the treatment with SM at 100 µg/mL, in which a significant increase ($p < 0.001$) in cell viability (%) of NHDF (normal human dermal fibroblasts) was observed.

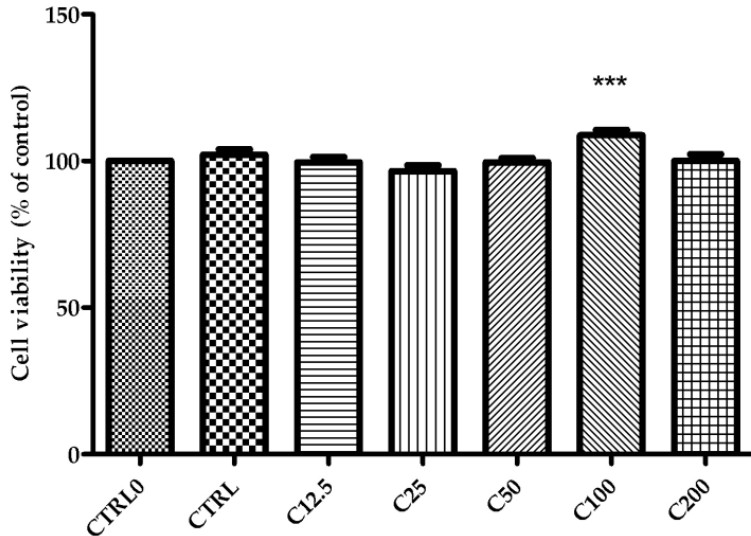

**Figure 3.** Effect of SM extract concentrations (between 12.5 µg/mL and 200 µg/mL) on viability of NHDF after 24 h. CTRL0—untreated cells; CTRL—positive control (cells treated with allantoin 50 µg/mL). Data are expressed as the mean ± SD from three individual experiments. *** $p < 0.001$ versus CTRL0.

*2.4. Evaluation of the Wound-Healing Effect of SM Extract Using In Vitro Scratch Assay*

2.4.1. Image Analysis

According to the image analysis, processed images were obtained that allowed the highlighting of the migration and proliferation of NHDF and the evolution of wound healing depending on the time and the treatment applied (Figure 4).

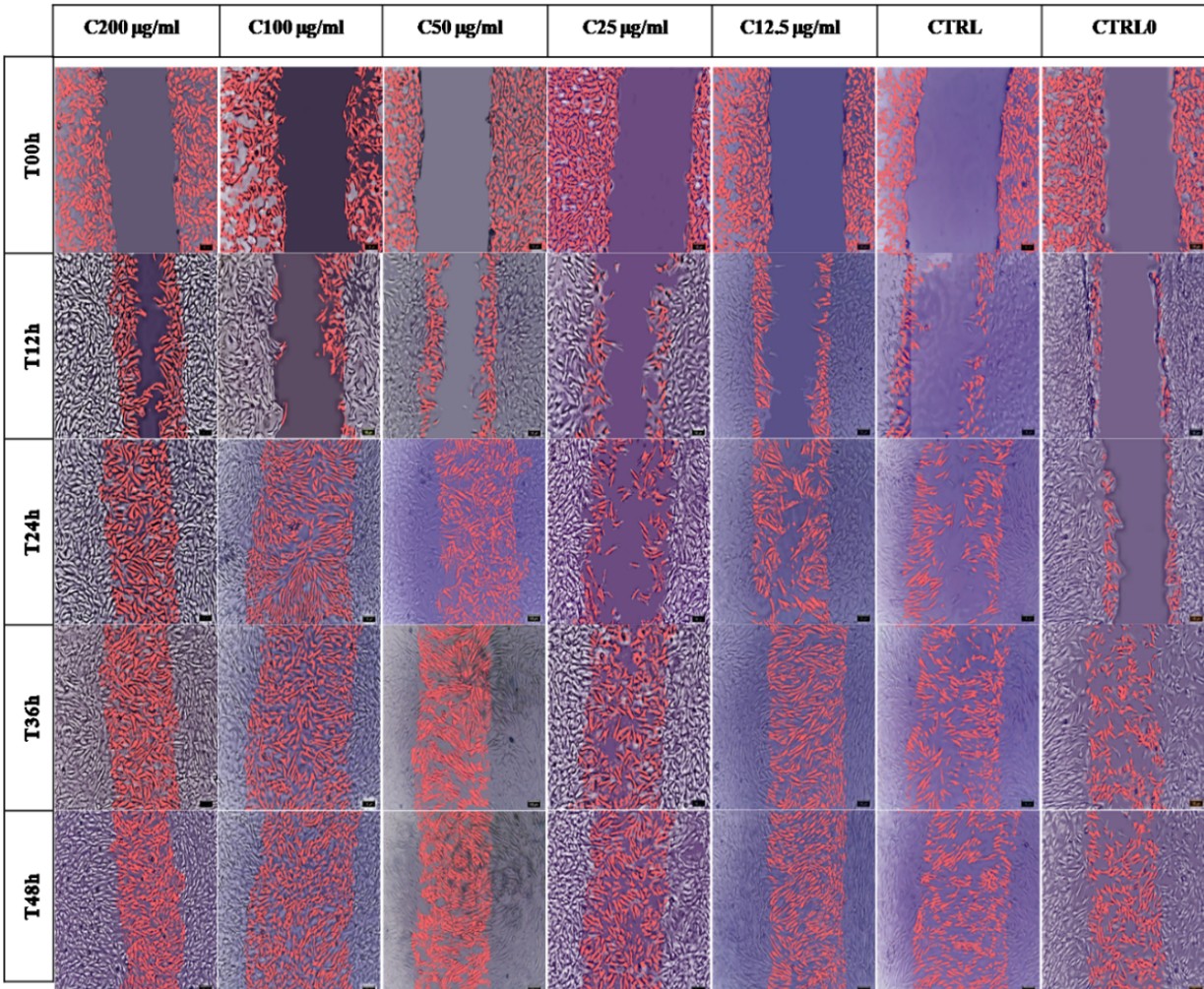

**Figure 4.** The evolution over time (0, 12, 24, 36, and 48 h) of cell wound coverage at different concentrations of *Stellaria media* (L.) Vill extract (range 200 µg/mL–12.5 µg/mL) compared to positive control (CTRL)—treated with allantoin 50 µg/mL and untreated wound control (CTRL0). Fibroblasts marked in pink are those that have mobilized into the wound. The scale of the processed images is 100 µm.

SM extract at a concentration of 50 µg/mL, 100 µg/mL, and 200 µg/mL indicated visually faster cell wound coverage at 24 h, while the concentrations of 25 µg/mL and 12.5 µg/mL appeared similar to the controls. The ability of fibroblasts to migrate into wounds in the presence of allantoin (positive control) was similar with the low concentrations of SM extract at 12 and 24 h but was slower compared with other SM concentrations. Based on the image analysis, it can be seen that at 36 and 48 h, the wound is closed depending on the treatments used.

2.4.2. ANOVA Statistics

Based on the image analysis, the following parameters were taken into account, which were statistically processed: relative wound width, wound area evolution, and normalized

cell density inside the wounds, depending on the time and the concentration of the SM extract (Supplementary Materials Tables S1–S3).

The values of relative wound width and area depended on the concentrations and the time of exposure to the SM extract. After 12 h, cell numbers in the wounded width were significantly different compared to the control, while at 48 h, no difference was recorded between samples (Tables S1 and S2).

The density of NHDF is a defining parameter in the wound-healing process, being the one that highlights the most effective treatment [18–20]. Figure 5 shows the cell density inside the wound depending on both the treatment applied and the time.

The normalized cell density factor is the one that best differentiates the effects of in vitro treatments. Figure 5 and Table S3 show the evolution of the cell density (%) compared with the control (untreated samples) for each treatment applied. Only the SM extract with a concentration of 100 µg/mL shows a significant percentage increase ($p = 0.05$) in cell density compared to the initial density (after 24 h). On the other hand, in the case of the SM extract of the highest concentration (200 µg/mL) and 50 µg/mL, an increase in cell density compared to the initial one is only observed after 36 h.

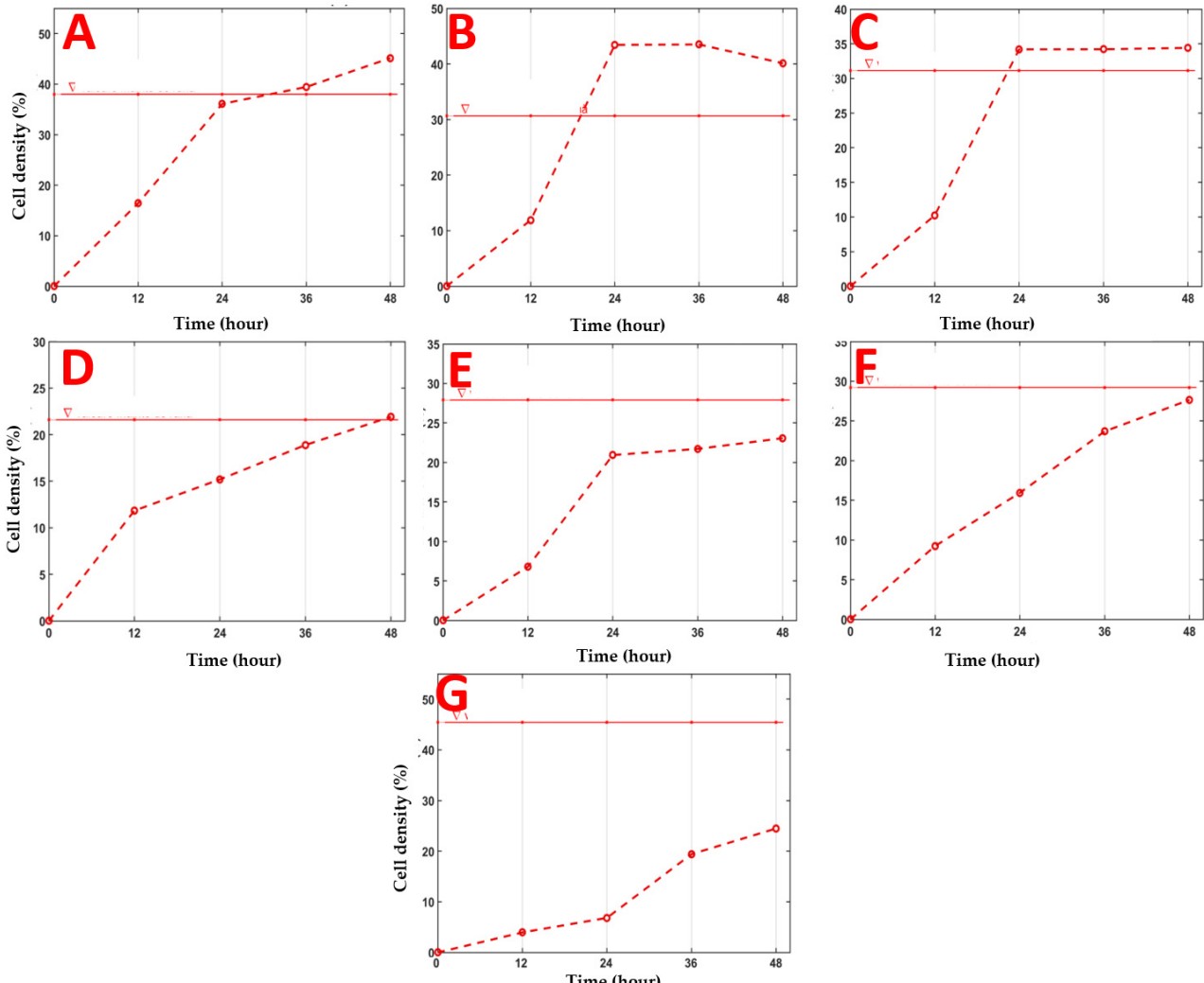

**Figure 5.** The evolution of the cell density parameter inside the wound depending on the applied treatment and time. The horizontal red line represents the initial density of the cells before the application of the "scratch" test, and the dotted red line represents the evolution in time of the cell density (%) depending on the applied treatment. *Stellaria media* (L.) Vill extract was applied in concentrations of: 200 µg/mL (**A**), 100 µg/mL (**B**), 50 µg/mL (**C**), 25 µg/mL (**D**), 12.5 µg/mL (**E**), Positive Control (CTRL), 50 µg/mL Allantoin (**F**), and control (CTRL0) (**G**), without any treatment.

The value of the $L^p$ norm—a quantitative parameter that allows the classification of samples taking into account the three normalized variables: coverage of the relative width of the wound (%), coverage of the wound area (%), and the normalized density of the cells inside the wound (%) for the removal of errors that may occur in univariate statistical analysis—was calculated with the data presented in Figure 6.

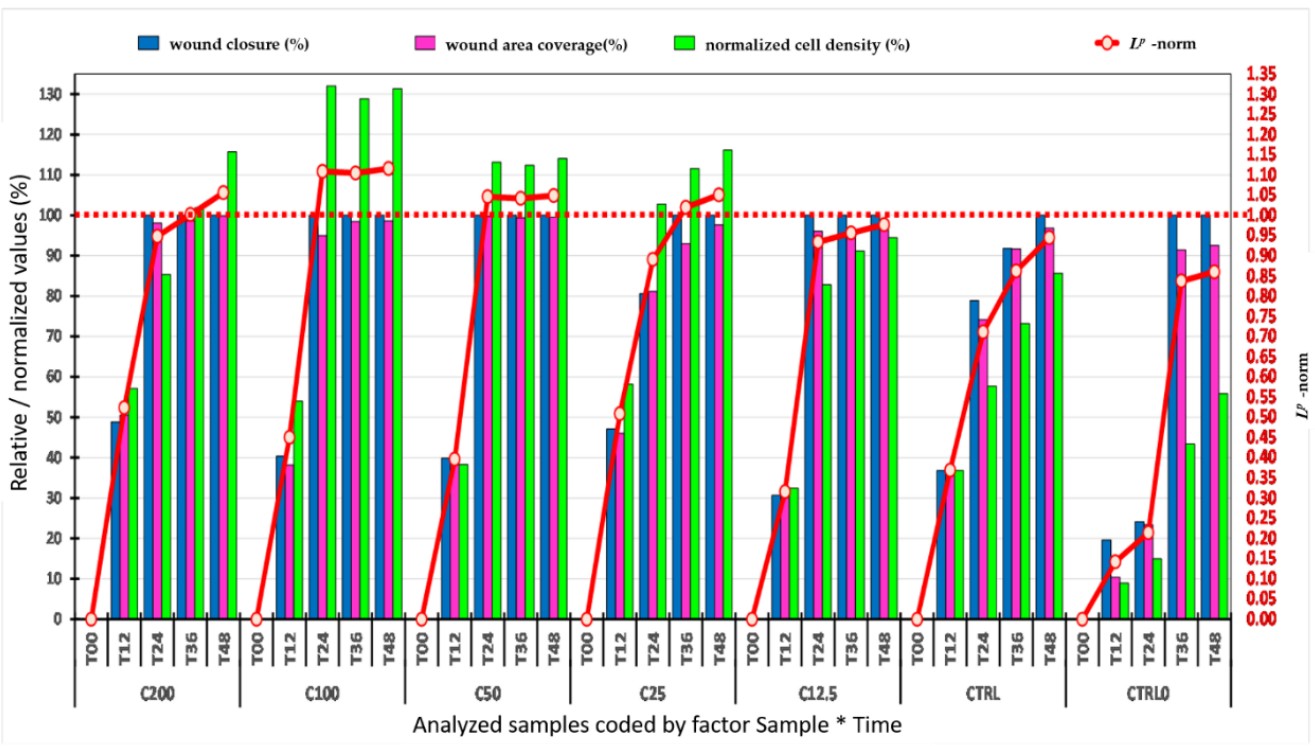

**Figure 6.** Evolution over time, of the average values of the widths (μm) of the wound, the areas (mm²) of the wound, of the normalized densities (%) of the cells inside the wound, and the $L^p$ norm, for all treatments. C200—12.5 μg/mL represents the *Stellaria media* (L.) Vill extract concentrations applied, CTRL—positive control (allantoin—50 μg/mL), CTRL0—control (without treatment), where no treatment was applied. T00–T48 represents the time expressed in hours.

Thus, all the results obtained by image and statistical analyses were represented in Figure 6 to highlight the most effective and fastest treatment that led to wound healing.

## 3. Discussion

In this work, the wound-healing potential of SM extract was tested in an in vitro wound-healing model using NHDF. In the first part of our study, the SM extract was characterized by phytochemical composition (total phenols and flavonoids), antioxidant capacity, and antimicrobial activity. The SM extract contains phenols and flavonoids, $17.19 \pm 1.32$ mg GAE/g dw and $7.28 \pm 1.18$ mg QE/g dw. Due to its bioactive compounds, SM extract possesses an antioxidant capacity, highlighted by FRAP assay and the electrochemical method (Table 1).

The results obtained in terms of content in bioactive compounds and antioxidant capacity are in accordance with data from the literature [8,14,21]. Table 3 shows the bioactive compounds identified in SM that contribute to the antioxidant, antimicrobial, and skin-repairing effects.

**Table 3.** Some biological activities of bioactive compounds identified in SM plants reported in the literature.

| Compounds Identified in SM | Biological Activities | | | References |
|---|---|---|---|---|
| | **Antioxidant** | **Antimicrobial** | **Wound Healing/ Skin Protection** | |
| Flavonoids | | | | |
| Apigenin | √ | - | √ | [22–25] |
| Genistein | √ | - | - | [26] |
| Vincenin-2 | √ | - | √ | [27,28] |
| Luteolin | √ | √ | - | [24,29–31] |
| Phenolic acid | | | | |
| Vanillic acid | √ | √ | - | [32,33] |
| Ferulic acid | √ | - | √ | [34,35] |
| Caffeic acid | √ | √ | √ | [36–40] |
| Chlorogenic acid | √ | √ | - | [41,42] |
| Vitamin | | | | |
| Vitamin C | √ | - | √ | [43,44] |
| Triterpenoid saponin | | | | |
| Gypsogenin | √ | - | - | [45] |
| Alkaloids | | | | |
| Stellarine A, B, C | - | √ | - | [46] |
| Dichotomine B, H, L | - | √ | - | [46] |

√: identified biological activity in SM; -: unidentified biological activity in SM.

Most flavonoids are known as antioxidants, along with vitamin C, due to their strong ability to donate electrons or hydrogen atoms [47,48].

SM plant extract can be characterized by the existence of a series of compounds such as flavonoids, phenolic acids, and triterpenoids [49]. Among the most representative flavonoids are derivatives of apigenin (7,2-di-O-hexosyl-6-C-hexosylapigenin, 6-C-3-(2-O-3-glucopyronosyl)-Glucopyronosyl-7-O-galactopyranosylapigenin, 6-C-13-(2-O-glucopyranosyl) glucopyranosyl-7-O-3-glucopyranosylapigenin and 6,8-C-diglycosylapigenin), genistein, and vicenin-2 [50]. From the flavonoid class, apigenin and Vicenin-2 are responsible for antioxidant and wound-healing effects (Table 3).

From the category of phenolic acids identified in the SM extract, vanillic acid, ferulic acid, chlorogenic acid, and caffeic acid are responsible for the antioxidant activity (Table 3), which was highlighted in our work by FRAP assay and the electrochemical method.

The SM plant has antioxidant properties based on its phytochemical composition, as pointed out by Oladeji et al., 2020, which states that the SM extract exhibited a high-level antioxidant capacity (76% for DPPH and 79% for FRAP) [14].

In addition to polyphenols, SM extract also has a considerable amount of vitamin C, significantly contributing to antioxidant activity [51].

Additionally, a recent study showed that gypsogenin, a triterpenoid saponin also identified in SM, has in vitro hepatoprotective and antioxidant activity [52].

The antioxidant capacity of SM extract is comparable to other species of the Caryophyllaceae family, such as *Silene gynodioca*, *Silene spergulifolia*, and *Silene swertiifolia* [21,53].

The peaks revealed by the DPV (Figure 2) can be assigned to the oxidation of the electroactive phenolic species from the extract [8,21,54–56]. The peak, which appeared at around 0.450 V, is due to the presence of ascorbic acid from the extract composition, as it was also revealed by other authors in comparable amounts [57].

The testing of the antibacterial activity of the SM extract was performed on Gram-negative (*E. coli*) and Gram-positive (*S. aureus*) bacteria, these being considered the main bacteria that are responsible for the production of infections in wounds [58]. According to research data, an important characteristic of the treatment applied on wounds is its antibacterial activity, which can asepticize the wound and determine the wound healing [59].

Table 2 shows the antimicrobial activity of the SM extract against the Gram-negative bacteria *E. coli*, which depends on the concentration used. It can be observed that the Gram-negative bacteria are sensitive to the SM extract at a concentration of 15 μg/mL, obtaining the inhibition diameter of 9.66 ± 0.58 mm. Higher antimicrobial activity of the SM extract was observed in the case of *S. aureus* (Gram-positive bacteria) compared with *E. coli*, which depends on the concentrations. Thus, according to Table 2, it can be stated that the SM extract has antimicrobial activity on Gram-positive bacteria; the largest inhibition diameter of 15.33 ± 0.58 mm was recorded on *S. aureus* at a plant extract concentration of 15 μg/mL.

The antimicrobial activity of extracts from the *Caryophyllaceae* family was evaluated on Gram-positive and Gram-negative bacteria. *Dianthus aryophyllus* extract was shown to have an antimicrobial effect on *Klebsiella pneumonia*, *Bordetella bronchiseptica*, and *Staphylococcus epidermidis* [60]. The antimicrobial activity of the *Caryophyllaceae* family was also demonstrated both on Gram-negative bacteria (*Proteus mirabilis* and *Escherichia coli*) and Gram-positive bacteria (*S. aureus*, *Bacillus cereus*, and *Listeria monocytogenes*) with a minimum inhibitory concentration value at 7.8 μg/mL and 15.6 μg/mL extract, respectively [61].

Different concentrations of SM extract have been tested on various bacteria, such as *S. aureus*, *E. coli*, *S. typhi*, *P. aureginosa*, and *K. pneumonia* și *B.cereus*, and it has been observed that the inhibitory effect increases with an increasing concentration. Furthermore, the antiviral activity against *Herpes simplex* and *Parainfluenza Virus* was demonstrated for species of the *Caryophyllaceae* family [62]. Luteolin and caffeic acid have antimicrobial effects against *S. aureus* and *E. coli*, respectively (Table 3).

SM extract was tested on NHDF cells to evaluate cell viability. After 24 h of incubation with different concentrations of SM extract, cell viability was not significantly different compared with the control in all treatments, except the SM of 100 μg/mL, where a significantly increased cell viability of NHDF was observed ($p < 0.001$) (Figure 3). Based on this result, all SM concentrations ranging between 12.5 μg/mL and 200 μg/mL were tested to evidence the healing effects.

The degree of proliferation and migration of fibroblasts in vitro was tested using the "scratch" method. This test is based on an assessment of the degree and rate of fibroblasts migration under the influence of plant extracts or drugs applied in order to "heal" an artificially produced wound in vitro [9].

Thus, by performing the scratch test, we wanted to identify which concentration of SM has the most intense healing and proliferative effects.

The image analysis in Figure 4 shows that the migration of cells in the wound for healing depends on the treatments applied and the time. It also allows the visualization of cells at time T00 h (their initial position—colored in white) and cells that have migrated to close the wound (colored in pink). For the validation of the image analysis, the univariate statistical analysis of the ANOVA type was applied. The parameters considered were: wound width (%), wound area (%), and normalized cell density (%).

Wound cell density is a parameter that must be considered to determine the most effective treatment applied [20]. Figure 5 shows the evolution of cell density inside the wound depending both on the treatment applied and time (T0 to 48 h) compared to the initial density. Treatment with C100 μg/mL SM extract causes the migration and rapid proliferation of cells, reaching a density higher than the initial one by 10% in 24 h. Additionally, the treatment with C50 extract μg/mL SM showed a higher increase in cell density than the initial one at 24 h. In contrast, in the case of treatment with the highest concentration of SM extract tested (C200 ug/mL), a high cell density is only observed after 36 h. Treatment with C25 ug/mL of SM extract reaches the initial cell density after 48 h.

Inside the wound, no cell density similar to the initial one was recorded at 48 h in the case of the treatment with the lowest concentration (12.5 μg/mL) and controls.

It can be stated that the application of the L$^p$ norm to the analyzed parameters was successful in delimiting the samples with the fastest and the most efficient wound-coverage processes [63].

Additionally, using the values of the L$^p$ norm, the most efficient variants of adding SM extract to the wound coverage can be pointed out.

After applying the scratch test on NHDF and testing the degree of stimulation of the SM extract on these cells, respectively, after applying the image analysis, the univariate statistical analysis, and the use of the L$^p$ norm, it can be stated that the C100 μg/mL concentration stimulated the migration of cells into the wound and healing with a cell density inside the wound higher than the initial density in the shortest time, compared to others applied (Figure 6).

Based on the statistical analysis, it was validated that the dose of 100 μg/mL of SM extract healed the wound produced in vitro after only 24 h and induced a cell density higher than the initial one. The SM extract has been shown to be more effective than allantoin, a compound known to have many properties associated with wound healing [64,65].

## 4. Materials and Methods

### 4.1. The Plant Material and Extract Preparation

The SM plant was collected from the city of Oradea, Romania (latitude-N 47.072782, longitude-E 21.910136) in April 2020. The identification of SM plant was made at the Department of Pharmaceutical Botany at the University of Oradea, Faculty of Medicine and Pharmacy. A specimen of the SM plant was kept and included in the herbarium of the Faculty of Medicine and Pharmacy Oradea, Romania, under the code: UOF05193.

Collected plant material was washed, completely dried at room temperature, and ground to form a powder. The powder of SM was macerated into 70% ethanol in a ratio of 1:20 (*w/v*). After 24 h and continuous stirring, in the dark, the mixture was filtered in vacuum, and the solvent was removed on a rotary evaporator (Heidolph Rotary Evaporator, Laborota 4000). The aqueous filtrate was freeze-dried (Christ Alpha 1–2 Ldplus lyophilizer) and stored at −20 °C until further use. For the evaluation of bioactive compounds, antioxidant capacity, and antimicrobial activity, the powder was freshly prepared by dissolving in distilled water to obtain a stock solution (1 mg/mL) that was further diluted into different concentrations.

### 4.2. Phytochemical Analysis

#### 4.2.1. Determination of Total Phenols Content

The total phenols content was determined using the Folin–Ciocâlteu method according to the literature [21,66,67]. Briefly, SM extract was mixed with freshly prepared Folin–Ciocâlteu reagent (1:10 dilution, *v/v*) and 7.5% $Na_2CO_3$ solution. The resulting mixture was incubated at room temperature in the dark for 2 h. The absorbance was measured at 765 nm using a Shimadzu mini UV-Vis spectrophotometer. The total polyphenol content of the extracts was expressed in mg equivalents of gallic acid (GAE)/g dry weight (dw).

#### 4.2.2. Determination of Total Flavonoid Content

The total flavonoid content of the extract was determined by the aluminum chloride colorimetric method described by Kim et al., 2003, and Tunde et al. [68,69], with little/minor modification. Briefly, an aliquot of SM extract (1 mL) was transferred to a 10.0 mL volumetric flask containing 4 mL distilled water. After 300 μL of 5% $NaNO_2$ was added to the flask, the mixture was allowed to stay 5 min. Then, 300 μL $AlCl_3$ 10% was added, and after 6 min, 2 mL of 1 M NaOH was added to the mixture, and the content of flask was completed with distilled water to obtain exactly 10.0 mL and thoroughly mixed. The absorbance was recorded at 510 nm versus blank. Quercetin was used as standard for

the quantification of total flavonoids, and the results were expressed as mg QE (quercetin equivalents)/g dw.

### 4.2.3. Determination of Antioxidant Capacity by FRAP Assay

The antioxidant capacity of the SM extract was determined by the FRAP (The Ferric-Reducing Antioxidant Power) assay according to the working technique described by Vicas et al. [66]. The principle of the FRAP method is based on the ability of the tested extract to reduce due to the antioxidant content of the ferriditripyridyl triazine complex (Fe (III)—TPTZ) to ferrous tripyridyl triazine (Fe (II)—TPTZ) in acidic pH [21,66,70,71]. The FRAP reagent was prepared using 300 mM acetate buffer (pH 3.6), ascorbic acid, 150 mg TPTZ (10 mM), and 150 μL HCl (40 mM), all homogenized with 50 mL distilled water. FRAP reagent was also prepared using acetate buffer, $FeCl_3$, TPTZ reagent in a ratio of 10:1:1 (*v/v/v*). The SM extract (100 μL) was homogenized with 500 μL FRAP reagents and 2 mL distilled water and then let to react in the dark for 1 h. The intensity of the blue coloration produced by the compound ferrous tripyridyl triazine (Fe (II)—TPTZ) was measured spectrophotometrically at 595 nm. Ascorbic acid with concentrations between 0.57 and 0.035 mM was used as standard for the calibration curve. The results were expressed in μg Ascorbic Acid equivalent (AAE)/mL.

### 4.2.4. Determination of Antioxidant Capacity by Electrochemical Method

The electrochemical experiments were performed by using a potentiostat (PGSTAT 128 N Autolab, Metrohm, Belgium) equipped with Nova 2.1.2 software. A three-electrode electrochemical cell was used with a glassy carbon electrode (GCE, 3 mm diameter, BAS) as working electrode, a Pt wire as counter electrode, and Ag/AgCl as reference electrode [72]. The GCE was polished with 2 μm diamond paste and subsequently rinsed with distilled water and ethanol before each measurement. The electrochemical behavior of the SM extract was analyzed by differential pulse voltammetry (DPV) with the following parameters: start potential −0.5 V, stop potential +1.5 V, step 0.01 V, modulation amplitude 0.05 V, modulation time 0.05 s, and interval time 0.1 s. Ascorbic acid equivalent antioxidant capacity (AEAC) was used as a standard for the evaluation of extract antioxidant capacity [57]. The calibration curve of ascorbic acid (in the range from $10^{-5}$ M to $5 \times 10^{-3}$ M) was performed by DPV, obtaining the following equation: $y = 0.1836 + 6537.5428 \times X$, ($R^2 = 0.998$).

### 4.3. Determination of Antimicrobial Activity

To test the antimicrobial activity of SM extract, agar-type Mueller–Hinton plates with a layer thickness of 90 mm were used, and the test method was the diffusion method [52,73]. Testing of the antimicrobial activity of the SM extract was performed on Gram-positive bacteria (*Staphylococcus aureus*) and Gram-negative bacteria (*Escherichia coli*) compared to different antibiotics (30 μg Doxycycline, 10 μg Gentamicin, 300 μg Nitrofurantoin, 5 μg Ciprofloxacin, 10U Penicillin).

The SM extract was applied in concentrations of 5 μg/mL, 10 μg/mL, and 15 μg/mL, and after 24 h at 37 °C, the diameter of the inhibition zone was compared to that produced by the antibiotics [73].

### 4.4. In Vitro Determination of the Healing and Proliferative Biological Effects of the SM Extract Using the Scratch Method

#### 4.4.1. Cell Culture and Treatments

Normal human dermal fibroblasts—adult (NHDF-Ad, CC-2511, Lonza, Basel, Switzerland) were cultured in sterile flasks (25 cm² surface) in a basal fibroblast growth medium (CC-3131, FGM, Lonza) with supplements (CC-4126, Lonza) according to the obtained producer protocol. The cell density used was 3500 cells/cm². The cell culture was maintained at 37 °C, enriched with 5% $CO_2$ of air atmosphere [74].

The stock solution of SM (500 mg/mL) was prepared by dissolving in Milli-Q water and then filtered by sterile filter. Prior to carrying out the treatments, a reference solution

of 5 mg/mL was made by diluting it with serum-free culture medium in order to reach final concentrations of 12.5 µg/mL, 25 µg/mL, 50 µg/mL, 100 µg/mL, and 200 µg/mL of SM extract.

### 4.4.2. Cell Viability Assay

The cell viability test was performed using trypan blue dye exclusion assay in an EVE Automatic cell counter (NanoEnTek Inc., Seoul, Korea) following standard protocol described by manufacturer. The use of the trypan blue dye method is an efficient and economical method of counting and testing cell viability according to the literature [75]. This test is based on the different permeability of the cell wall. Thus, in the case of dead cells, the dye will penetrate through the cell wall, and in the visual field of cell counter, they will be displayed in blue [75]. Living cells are visualized translucent [75,76].

NHDF-Ad cells were seeded into 24-well plates at $1 \times 10^4$ cells/well and maintained at $37\,^\circ$C with 5% $CO_2$ for 24 h to obtain complete attachment. Then, the cells were treated with SM extract at doses of 12.5 µg/mL, 25 µg/mL, 50 µg/mL, 100 µg/mL, and 200 µg/mL for 24 h. The untreated cells were the control (CTRL0), and allantoin-treated cells of 50 µg/mL represent the positive control (CTRL). After, the adhered cells were trypsinized, and the 10 µL of cells sample and 10 µL trypan blue stain 0.4% was mixed. Then, 10 µL of mixture was applied to an EVE cell counting slide, and live cells were determined using the EVE automatic cell counter. The cell viability was calculated according to Formula (1). The results were expressed as percentage of cell viability of treated cells against control. The measurements were performed in triplicate, and the data are represented as mean ± standard deviation (SD) ($n = 3$).

$$\text{Cell viability (\%)} = \frac{\text{number of live cells}}{\text{number of total cells}} \times 100 \tag{1}$$

### 4.4.3. Scratch Assay

The migration and healing speed of wounds produced in vitro was tested by the scratch method. The cell density used was $2 \times 10^4$, the cells being placed in sterile 6-well plates. Cells prepared for the scratch method were kept in the incubator at $37\,^\circ$C and 5% $CO_2$ until confluence (48 h); then, the wound was simulated *in vitro*. The fibroblast monolayer was scratched with a sterile 100 µL plastic pipette tip. The cellular debris was removed by washing with HEPES buffer [74]. Subsequently, the SM extract was applied directly on the scratch in different concentrations of 12.5 µg/mL, 25 µg/mL, 50 µg/mL, 100 µg/mL, and 200 µg/mL. Untreated scratch was used as control (CTRL0). The allantoin solution (50 µg/mL) placed in scratch (CTRL) was considered as a positive control (CTRL) according to the literature [74,77,78]. After the treatments, the fibroblasts were incubated at $37\,^\circ$C and 5% $CO_2$ for different time intervals (12, 36, and 48 h).

The microscopic evaluation was performed with the Olympus XC30 optical microscope with $10\times$ objective in phase contrast, at 0, 12, 36, and 48 h. The images were then analyzed to determine the rate of cell migration.

The healing speed of scratches after applying different treatments was monitored at different times (T12, T24, T36, and T48) and compared to the initial time (T00).

### 4.5. Statistical Analysis

#### 4.5.1. Image Analysis

The images obtained with the microscope, which represent the evolution of the wound coverage with fibroblasts, were processed in order to quantitatively evaluate the process of migration and proliferation inside the wound of fibroblast cells. The proposed quantitative sizes, at the raw level, are the width of the wound, the area of the wound, and the density of cells inside the wound.

The captured images have a size of $2080 \times 1544$ pixels and a resolution of 76 dpi, and are in JPEG format. Because dermal fibroblast cells are optically semitransparent, these original images have very poor contrast between the fibroblast cells and the area delimited

by the wound. As a result, in order to achieve the purpose described above, the original images must be processed to obtain a contrast high enough to differentiate the cells from the wound area and the background.

### 4.5.2. Univariate Statistical Analysis (ANOVA)

The univariate statistical analysis involved the application of the bifactorial ANOVA method to the three parameters analyzed. The proposed factors are the test with the levels: C200, C100, C50, C25, C12.5, CTRL, and CTRL0 (the samples named according to the concentration of the extract, the positive control samples with allantoin and control without any addition), and time with the levels: T00, T12, T24, T36, and T48 (the samples named according to the number of hours from the start of the experiment for each sample) and the interaction factor Sample * Time. The interaction factor has the levels: C200_T00, C200_T12, C200_T24, C200_T36, C200_T48, C100_T00, C100_T12, C100_T24, C100_T36, C100_T48, C50_T00, C50_T12, C50_T24, C50_T00, C25T, C25T, C50T, C50T C12.5_T12, C12.5_T24, C12.5_T36, C12.5_T48, CTRL_T00, CTRL_T12, CTRL_T24, CTRL_T36, CTRL_T48, CTRL0_T00, CTRL0_T12, CTRL0_T24, CTRL0_T36, and CTRL0_T48.

The quantitative parameters analyzed are relative width (%), area (%), and normalized cell density (%) inside the wound [79,80].

The assessment of wound coverage by relative width (%) was calculated for each sample at each time point by using Relation (2):

$$\mathrm{d} = \frac{\text{The area of the cells inside the wound\_sample } (\mathrm{mm}^2) \text{ at time t}}{\text{Wound\_sample\_sample}(\mathrm{mm}^2) \text{ at time t} = \text{T00}} \times 100 \qquad (2)$$

The assessment of wound coverage by area (%) was calculated for each sample at each time point by using Relation (3):

$$\text{Area } (\%) \text{ sample, at time t} = \frac{\text{Wound area\_sample } (\mathrm{mm}^2) \text{ at time t}}{\text{Wound area\_sample } (\mathrm{mm}^2) \text{ at time t} = \text{T00}} \times 100 \qquad (3)$$

From a medical point of view, the density of cells inside the wound is the most relevant quantitative parameter in wound treatments (Figure 4). Wound cell density (d) was calculated by the ratio of wound cell density (%) for each sample and at different times (T00 h, T12 h, T24 h, T36 h, and T48 h) and wound area (mm$^2$) for each sample at the time T00 h.

The assessment of wound coverage by normal density (%) was calculated for each sample at each time point by using Relation (4):

$$\mathrm{d} = \frac{\text{The area of the cells inside the wound\_sample } (\mathrm{mm}^2) \text{ at time t}}{\text{Wound\_sample\_sample}(\mathrm{mm}^2) \text{ at time t} = \text{T00}} \times 100 \qquad (4)$$

### 4.5.3. L$^p$ Method

If the experimental design allows the recalculation of quantitative parameters to present the same range of variation, a quantitative quantity called the L$^p$ norm can be calculated with $p > 2.00$. This size will characterize the multivariate analyzed samples.

If Np, the quantitative parameters $V_{i,\ i=\overline{1,Np}}$, are present in the experimental design, the norm L$^p$ is defined by Relation (5) [63,81,82]:

$$L^p = \sqrt[p]{\frac{\sum_{i=1}^{Np} V_i^p}{Np^* [\max(V_i)]^p}} \qquad (5)$$

In this way, practically, by calculating the L$^p$ norm, a multivariate score is obtained and used for the classification of the analyzed samples.

## 5. Conclusions

This study investigated the antioxidant capacity and antimicrobial effect of SM extract and the potential wound-healing effect of SM in NHDF cells. The antioxidant capacity of SM extract highlighted by the FRAP assay and the electrochemical method is mainly due to the content of phenols and flavonoids. Additionally, the SM extract showed a more pronounced antimicrobial effect on Gram-positive bacteria (*S. aureus*) compared with Gram-negative (*E. coli*) ones at a level of 15 μg/mL. Our findings demonstrate, for the first time in the literature, the wound-healing potential of SM extract on NHDF using the in vitro scratch method. The combination of microscopic evaluation with image analysis, univariate statistical analysis, and $L^p$ norm multivariate score revealed that the SM extract of 100 μg/mL determines the highest cell migration and proliferation, reaching a density higher than the initial one by 10% in 24 h. The treatment with SM extract at 100 μg/mL resulted in wound-healing in vitro in a shorter time (24 h) compared with the positive control (50 μg/mL Allantoin). Further studies are needed to identify the SM compounds responsible for the proliferation and migration of fibroblast and elucidate the mechanisms underlying wound-healing mechanisms.

**Supplementary Materials:** The following are available online at https://www.mdpi.com/article/10.3390/app112311526/s1, Table S1. Effect of SM extract (12.5–200 ug/mL) on the migratory and proliferative activity of fibroblasts, expressed as width of the wound in the scratch assay after 0, 12, 24, 36, and 48 h (T00, T12, T24, T36, and T48, respectively). CTRL means positive control—50μg/mL Allantoin, CTRL0 means control, (untreated wound), C200 means treatment with 200 μg/mL SM extract, C100 means treatment with 100 μg/mL SM extract, C50 means treatment with 50 μg/mL SM extract, C25 means treatment with 25 μg/mL SM extract, C12.5 means treatment with 12.5 μg/mL SM extract. Data are expressed as relative ratio of cell numbers in the wounded width compared to the control for the analyzed samples (mean ± standard deviation, N = 3). Results are presented for the Sample factor * Time from bivariate analysis (ANOVA), Table S2. Effect of SM extract (12.5–200 ug/mL) on the migratory and proliferative activity of fibroblasts, expressed as area of the wound in the scratch assay after 0, 12, 24, 36, and 48 h (T00, T12, T24, T36, and T48, respectively). CTRL means positive control—Allantoin 50μg/mL, and CTRL0 means control (untreated wound), C200 means treatment with 200 μg/mL SM extract, C100 means treatment with 100 μg/mL SM extract, C50 means treatment with 50 μg/mL SM extract, C25 means treatment with 25 μg/mL SM extract, C12.5 means treatment with 12.5 μg/mL SM extract. Data are expressed as the area of the wound compared to the control for the analyzed samples (mean ± standard deviation, N = 3). Results are presented for the Sample factor * Time from bivariate analysis (ANOVA), Table S3. Effect of SM extract (12.5–200 ug/mL) on the migratory and proliferative activity of fibroblasts, expressed as normalized density of cells of the wound in the scratch assay after 0, 12, 24, 36, and 48 h (T00, T12, T24, T36, and T48, respectively). CTRL means positive control—Allantoin 50μg/mL, and CTRL0 means control (untreated wound), C200 means treatment with 200 μg/mL SM extract, C100 means treatment with 100 μg/mL SM extract, C50 means treatment with 50 μg/mL SM extract, C25 means treatment with 25 μg/mL SM extract, C12.5 means treatment with 12.5 μg/mL SM extract. Data are expressed as normalized density of cells in the wound compared to the control for the analyzed samples (mean ± standard deviation, N = 3). Results are presented for the Sample factor * Time from bivariate analysis (ANOVA).

**Author Contributions:** Conceptualization, S.I.V. and S.C.; methodology, F.M., L.F., V.L. and A.R.M.; software, A.C.T., A.M.V. and V.L.; validation, M.G., M.Z. (Marcel Zdrinca) and L.D.; formal analysis, M.Z. (Mihaela Zdrinca) and A.M.V.; investigation, F.M.; resources, A.A., M.Z. (Marcel Zdrinca) and L.D.; writing—original draft preparation, F.M. and M.G.; writing—review and editing, S.I.V. and S.C.; visualization, V.L. and A.R.M.; supervision, V.L.; project administration, P.P.; funding acquisition, A.A. and M.Z. (Mihaela Zdrinca), M.Z. (Marcel Zdrinca), M.G., P.P. and L.D. All authors have read and agreed to the published version of the manuscript.

**Funding:** This research received no external funding.

**Informed Consent Statement:** Not applicable.

**Data Availability Statement:** Data are available in a publicly accessible repository.

**Acknowledgments:** The authors of this article thank Angela Antonescu for the acquisition of the cell line. This study is part of a complex project in which various plant extracts (*Stellaria media*, *Ocimum basilicum,* and *Trifolium pratense*) were tested on human dermal fibroblasts using the scratch assay. Statistical processing and presentation of results were applied for each extract, the data being presented in different papers.

**Conflicts of Interest:** The authors declare no conflict of interest.

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
