# Peer review of "Evaluation of In Vitro Wound-Healing Potential, Antioxidant Capacity, and Antimicrobial Activity of Stellaria media (L.) Vill"

_applsci, doi:10.3390/app112311526_

Round 1

Reviewer 1 Report

In the present study titled “Evaluation of in vitro wound healing potential, antioxidant capacity and antimicrobial activity of Stellaria media (L.) Vill” it was aimed to investigate the effect of SM extract on the wound healing process. This study has some deficiencies and should be revised according to the points mentioned below:

  • Too much detailed information is exhibited in the introduction part, and the paragraphs are disconnected from each other. Therefore, this section should be revised.
  • Since this article is not a methodological study, it is not appropriate to give information about the method in the introduction part. Therefore, it is recommended to remove the part related with the scratch method from the introduction section.
  • Abbreviations should be explained where they first appear in the text (such as AEAC in the result 2.1.)
  • In the result part (2.1.) the authors have written that “As shown in Table 1, the values of antioxidant capacity are different, because the methods’s mechanisms are different”. However this sentence contains a comment related with obtained data and this is not a result sentence. This should be revised.
  • In the part “2.4.2. Anova Statistics”, the sentence begining with “The density of NHDF is a defining parameter in the wound healing process...” should be supported by the references.
  • It is recommended to explain the scratch test results in more detail.
  • If a cytotoxicity assessment has been performed, the purpose of this assay should be stated in the material method section. In addition, it should be explained more clearly which formulation is used in the calculation of relative cell proliferation.

Author Response

Dear Reviewer 1,

Thank you for your suggestions that significantly improve our article

  • Too much detailed information is exhibited in the introduction part, and the paragraphs are disconnected from each other. Therefore, this section should be revised.

The Introduction part has been completely revised and is marked in red in the manuscript.

  • Since this article is not a methodological study, it is not appropriate to give information about the method in the introduction part. Therefore, it is recommended to remove the part related with the scratch method from the introduction section.

The introduction has been modified and we remove the part related with the scratch method.

  • Abbreviations should be explained where they first appear in the text (such as AEAC in the result 2.1.)

This problem has been solved, in the text the changes are in red.

  • In the result part (2.1.) the authors have written that “As shown in Table 1, the values of antioxidant capacity are different, because the methods’s mechanisms are different”. However, this sentence contains a comment related with obtained data and this is not a result sentence. This should be revised.

We removed the phrase from part 2.1.

  • In the part “2.4.2. Anova Statistics”, the sentence begining with “The density of NHDF is a defining parameter in the wound healing process...” should be supported by the references.

The references data were added as required. In the text are included as 18, 19, 20.

  1. Chung, H.H.; Bellefeuille, S.D.; Miller, H.N.; Gaborski, T.R. Extended Live-Tracking and Quantitative Characterization of Wound Healing and Cell Migration with SiR-Hoechst. Exp Cell Res 2018, 373, 198–210, doi:10.1016/j.yexcr.2018.10.014.
  2. Browning, A.P.; Jin, W.; Plank, M.J.; Simpson, M.J. Identifying Density-Dependent Interactions in Collective Cell Behaviour. J R Soc Interface 2020, 17, 20200143, doi:10.1098/rsif.2020.0143.
  3. Antonescu (Mintaș), I.A.; Antonescu, A.; Miere (Groza), F.; Fritea, L.; Teușdea, A.C.; Vicaș, L.; Vicaș, S.I.; Brihan, I.; Domuța, M.; Zdrinca, M.; et al. Evaluation of Wound Healing Potential of Novel Hydrogel Based on Ocimum Basilicum and Trifolium Pratense Extracts. Processes 2021, 9, 2096, doi:10.3390/pr9112096.
  • It is recommended to explain the scratch test results in more detail.

We included in text more explication (in red color) of the scratch test results.

  • If a cytotoxicity assessment has been performed, the purpose of this assay should be stated in the material method section. In addition, it should be explained more clearly which formulation is used in the calculation of relative cell proliferation.

The formula for calculation of cell viability was included in the Materials and Methods Chapter. Based on the results of cell viability (%) of treated cells with different SM concentrations we didn’t use another method to evaluate cytotoxicity.

Reviewer 2 Report

The authors investigate an extract of Stellaria media in the context of composition, antioxidant and anti-microbial activity and enhanced scratch closure in a HDF mononlayer. These in vitro activities may be of benefit for improved wound healing. The authors use acceptable methods to adress their questions,  and the obtained data are presented in a more or less understandable manner.

Concerns: 

  • In the cell viability assay, a positive control is missing
  • In some aspects, the data appear preliminary/of limited scope and do not score too high when it comes to novelty. For example, it is not suprising that phenolic components show antixidant activity.  
  • The rather complicated statistical analysis (with Lp etc) seems to be somewhat exaggerated for a simple in vitro assay which only can give first indications on potential pro-migratory/pro-proliferative activity of the extract. Moreover, the authors did not clearly indicate the number of performed independent experiments or provide any reference for their claim, that the cell density in the wound is the best predictor for efficient wound healing.
  • Instead of dwelling on the extensive statstical analysis of a first in vitro assay, and in order to increase the impact of the study, the authors should invest more effort in order to  get more information either on the responsible active ingredient or the underlying mode of action for the faster scractch closure
  • The outlook to use SM as starting material for a new wound healing therapeutic (as e.g. mentioned in the abstract)  is very speculative based on the presented data set
  • Another round of language editing is necessary as there are couple of clumpsy passages in the text (e.g line 290-300)
  • Also, a total number of 14 authors for a comparably limited data set should be seriously reconsidered.  

Author Response

Dear Reviewer 2,

Thank you for your suggestions that significantly improve our article

  • In the cell viability assay, a positive control is missing

A positive control (CTRL) was included in the Figure 3.

  • In some aspects, the data appear preliminary/of limited scope and do not score too high when it comes to novelty. For example, it is not surprising that phenolic components show antioxidant activity.

Some aspects of the manuscript indicate novelty aspects. From the data we have so far, this is the first time that the wound healing effect for the Stellaria media has been highlighted. This plant is not very well characterized from a phytochemical point of view, and in the future we will try to identify the compound / compounds responsible for this effect. Usually, the antioxidant activity is evaluated by SET or HAT methods, and in this manuscript, we presented an electrochemical method to highlight the antioxidant capacity, which is another aspect of novelty.

  • The rather complicated statistical analysis (with Lp etc.) seems to be somewhat exaggerated for a simple in vitro assay which only can give first indications on potential pro-migratory/pro-proliferative activity of the extract. Moreover, the authors did not clearly indicate the number of performed independent experiments or provide any reference for their claim, that the cell density in the wound is the best predictor for efficient wound healing.

The Lp norm is a quantitative parameter that gives statistical calculations a high accuracy. We chose to perform a detailed statistical analysis in order to validate the results, these being obtained by comparing three independent repetitions of samples.

The references that confirm that the cell density parameter is defining in the wound healing process have been introduced in the text. These are:

18.Chung, H.H.; Bellefeuille, S.D.; Miller, H.N.; Gaborski, T.R. Extended Live-Tracking and Quantitative Characterization of Wound Healing and Cell Migration with SiR-Hoechst. Exp Cell Res 2018, 373, 198–210, doi:10.1016/j.yexcr.2018.10.014.

19.Browning, A.P.; Jin, W.; Plank, M.J.; Simpson, M.J. Identifying Density-Dependent Interactions in Collective Cell Behaviour. J R Soc Interface 2020, 17, 20200143, doi:10.1098/rsif.2020.0143.

20.Antonescu (Mintaș), I.A.; Antonescu, A.; Miere (Groza), F.; Fritea, L.; Teușdea, A.C.; Vicaș, L.; Vicaș, S.I.; Brihan, I.; Domuța, M.; Zdrinca, M.; et al. Evaluation of Wound Healing Potential of Novel Hydrogel Based on Ocimum Basilicum and Trifolium Pratense Extracts. Processes 2021, 9, 2096, doi:10.3390/pr9112096.

  • Instead of dwelling on the extensive statistical analysis of a first in vitro assay, and in order to increase the impact of the study, the authors should invest more effort in order to get more information either on the responsible active ingredient or the underlying mode of action for the faster scratch closure.

Based on data from the literature on the phytochemical composition of MS, we marked compounds that have a wound healing effect (Table 3). Without experimental data we can only assume that there is a synergistic effect between the compounds present in the extract, for example between apigenin and Vincenin-2. In the future we will try to elucidate the mechanisms of wound healing  by MS.

  • The outlook to use SM as starting material for a new wound healing therapeutic (as e.g mentioned in the abstract) is very speculative based on the presented data set

             The abstract was modified with an emphasis on the results obtained in this study.

  • Another round of language editing is necessary as there are couple of clumsy passages in the text (eg. line 290-300)

The paragraph on line 290-300 has been revised and also we corrected language editing throughout the manuscript.

  • Also, a total number of 14 authors for a comparably limited data set should be seriously reconsidered.  

Each author taken part in this study according to "Author's Contributions"